# The Impact of the Environment on the Quality of Life and the Mediating Effects of Sleep and Stress

**DOI:** 10.3390/ijerph17228529

**Published:** 2020-11-17

**Authors:** Katherine Ka Pik Chang, Frances Kam Yuet Wong, Ka Long Chan, Fiona Wong, Hung Chak Ho, Man Sing Wong, Yuen Shan Ho, John Wai Man Yuen, Judy Yuen-man Siu, Lin Yang

**Affiliations:** 1School of Nursing, The Hong Kong Polytechnic University, Hung Hom, Kowloon, Hong Kong, China; katherine.chang@polyu.edu.hk (K.K.P.C.); janice.ys.ho@polyu.edu.hk (Y.S.H.); john.yuen@polyu.edu.hk (J.W.M.Y.); l.yang@polyu.edu.hk (L.Y.); 2Department of Land Surveying and Geo-Informatics, The Hong Kong Polytechnic University, Hung Hom, Kowloon, Hong Kong, China; kalong.chan@polyu.edu.hk (K.L.C.); ls.charles@polyu.edu.hk (M.S.W.); 3School of Optometry, The Hong Kong Polytechnic University, Hung Hom, Kowloon, Hong Kong, China; fionawong1@gmail.com; 4Department of Urban Planning and Design, The University of Hong Kong, Pokfulam, Hong Kong, China; hcho21@hku.hk; 5Department of Applied Social Sciences, The Hong Kong Polytechnic University, Hung Hom, Kowloon, Hong Kong, China; judy.ym.siu@polyu.edu.hk

**Keywords:** quality of life, environment, stress, sleep

## Abstract

(1) *Background*: Environment is an independent factor that affects one’s quality of life (QoL), where studies suggest that health behaviours also affect one’s quality of life. The purpose of the present study was to examine the association between environmental conditions and QoL and how individual health behaviours affect this association. (2) *Methods*: Participants aged 20 or above were recruited from 11 tertiary planning units in the central part of Kowloon. These tertiary planning units were selected as they represented the overall living environment in Hong Kong, with a mix of the poorer urban areas alongside relatively affluent districts. A mediation analysis was implemented using multiple linear regressions to examine the effects of environmental conditions on QoL. (3) *Results*: Of the 607 eligible participants included for analysis, 390 were female and 217 were male, with a mean age of 47.4 years. Living within 500 m of a green space area had benefits on the physical aspect of QoL and physical activity but no effect on the psychological aspect of QoL. Moderate satisfaction with public spaces affected QoL positively. In contrast, less satisfaction with public spaces affected QoL negatively in both physical and psychological aspects through the mediating effect of stress. Poor environmental quality affected all domains of QoL negatively through the mediating effects of increased stress and poor sleep. (4) *Conclusions*: Environment is an important factor that affects individuals’ overall well-being. The interaction between environmental conditions and individual variables, especially perceived stress and sleep, is extremely important when assessing its impact on QoL. The findings of this study support the importance of individual stress and sleep in mediating the relationship between the environment and QoL for health. Further studies should be conducted to include objective measurements, such as those of cortisol levels for stress and physical fitness tests.

## 1. Introduction

Quality of life (QoL) is a broad multidimensional concept that is significant as a desired health outcome. It represents the expectation and concern for one’s own health and life, including both positive and negative aspects of QoL in the context of the culture and value systems [1]. The 36-item Short-Form Health Survey (SF-36) is a widely used health-related QoL tool that is aimed at detecting a subjective expression of health status with eight health concepts in the two main dimensions of physical and mental health [2]. However, these dimensions focus on a disease’s effect on specific functional aspects but do not address the QoL that is embedded in a cultural, social and environmental context [3]. The World Health Organization (WHO) thus initiated the development of the World Health Organization Quality of Life 100 (WHOQOL 100) to consider people’s overall well-being. The WHO Quality of Life Brief Scale (WHOQOL-BREF), a short form of the WHOQOL100, was field-tested, which resulted in the identification of a four-domain structure: physical health, psychological health, social relationships and the environment in general. These four constructs were found to have moderate correlations but had adequate discriminant validity, reflecting the different measures they each represent [4].

The dimensions of QoL for health traditionally include physical, psychological and social aspects, but in recent years, the environment has been identified as another important dimension of life [5]. Green space in urban areas constitutes an environmental determination to improve both physical and psychological health, enhance the quality of living and resilience in urban areas and promote sustainable lifestyles among urban residents [6]. It provides access to nature and scenic beauty in urban areas, which may influence physical health [7], psychological health [8,9] and environmental health [10]. The distance to green space has been emphasised as an important factor that affects health. A Danish study based on a national survey found that residents living closer than 1 km to green space had higher mean scores on all eight subscales of the SF-36 than those living more than 1 km away [11]. A British study that included all residents appearing in the Census found that the percentage of green space was associated with better health in general, although the effects were not significant among a group with higher income and those living in suburban areas [12]. However, a German study did not find any association between green space and health-related QoL [13]. Other studies revealed that the quantity of green space is associated with health. A greater expanse of forest in rural and urban green spaces was associated with fewer mental health issues [14]. A systematic review supported the positive association of the quantity of green space with psychological health, although the evidence for the relationship between green spaces and general health is less strong [15].

Public spaces in neighbourhoods with man-made recreational resources, such as walking trails, cycling areas and swimming pools, are advantageous to physical activity [5] and contribute to overall feelings about the community [16] and QoL [17,18]. A survey was conducted among Australian adults and found that environmental attributes, including access to parks, bicycle and walking trails and the presence of green space, were positively related to physical and psychological health [18]. Similarly, a Dutch survey found that environmental qualities from buildings, noise and traffic were strongly related to all four domains of QoL, as measured by the WHOQOL-BREF [17]. Perception of the public open space and built environment are essential constituents in the evaluation of the neighbourhood, health [19] and QoL [20]. Our earlier study, however, found that satisfaction with the neighbourhood environment was only significantly related to psychological QoL [21]. Residents who reported perceiving moderate satisfaction with the public space in the neighbourhood had a significantly higher psychological QoL.

Previous studies reported possible mediators that might explain the relationship between the environment and QoL, including socioeconomic status and individual health behaviours [22,23,24]. Green spaces appear to have a stronger effect on health among lower socioeconomic classes within a Dutch population [25]. Mitchell and Popham [12] argued that the income-related inequality in health of the lower socioeconomic subpopulation could be reduced through greater exposure to green space. A sub-analysis was conducted among a Dutch data set from a population survey and found that the green space effect on health was stronger among the elderly and homemakers [26]. These two groups were suggested to be more confined and thus more exposed to the local environment. Apart from socioeconomic status, several mediator variables have been proposed that can clarify the environmental effect on QoL through individual health behaviours. Gong et al. [27] found that an increased percentage of green space within a 400-metre area was significantly associated with more participation in physical activity among a group of elderly men in the U.K., but few differences in the levels of physical activity were found in a New Zealand population [28]. People who engaged in physical activity and who had greener households reported less stress [29] and improved physical and psychological health [29,30,31].

Stress is a subjective response to the perception of external influences, including environmental factors [32]. Those who live closer to green spaces were found to have lower odds of experiencing stress, as measured by the Perceived Stress Scale [11]. These results supported those found by Maas et al. [25]. Comparison between people living within a 3 km and 1 km radius from green space found that both the elderly and the youth who live within 1 km of it perceived better general health. Living closer to green space is associated with decreased cortisol levels and stress [33] and with better QoL [11]. Individuals with more perceived stress were found to have a higher chance of poor sleep quality, but the adverse effect of stress on sleep quality could be attenuated by more green space [34]. Perceived stress may also serve as a mediator for health. Studies suggest that stress can mediate health outcomes by reinforcing individual health behaviours [26,35,36]. Sleep has also been associated with neighbourhoods that have green space. A Canadian population-based survey explored the psychological well-being and QoL associated with sleep [37]. They found that poor sleep was linked to impaired quality of life and increased stress. Environmental features, such as light, noise and traffic, can affect perceived sleep quality, impaired mood [38] and produce longer reaction times [39]. Cote and colleagues found that reaction times get longer among individuals with sleep deprivation. People seem to be compensating for longer reaction times by using more mental effort, as measured by a high-frequency electroencephalogram [39], thus impairing mood and psychological well-being. People who live in an environment that has 80% or more green space reported longer sleep duration at night [40]. Other individual health behaviours, such as smoking, drinking [16,21] and fruit and vegetable consumption [41], are considered to have an impact on QoL [16].

According to Marans and Mohai’s theoretical model [42], health and QoL may be linked not only with leisure resources but also environmental conditions [43]. The model hypothesised that the perceptions of environmental and urban amenities will influence peoples’ satisfaction, physical health and their use of such amenities. Environmental amenities include the quality of the ambient environment (air, water, noise and hazardous waste) and natural recreation resources. Urban amenities include man-made recreational resources (walking trails, swimming pools and cycling areas), cultural resources (sports teams, cinemas and galleries), health services and facilities, public space and public transport [44]. The use or non-use of natural or man-made recreational resources by an individual is associated with physical health [44] and individual health behaviours, such as physical activity [45]. The model provides opportunities to explore the relationship between environmental conditions and QoL. In this study, we proposed that the effect of environmental conditions is not only associated with but also interacts with individual health behaviours and QoL. These relationships have not previously been well studied; therefore, the aim of this study was to examine environmental conditions and individual health behaviours as mediating factors that affect QoL.

## 2. Materials and Methods

### 2.1. Eligible Participants and Recruitment

Participants were recruited from 11 tertiary planning units (TPUs) in the centre of Kowloon. These TPUs were selected as they represented the overall living environment in Hong Kong, with older, poorer urban areas alongside relatively affluent districts. A sample size of 598 provided a precision range of 4% from the true values at a 95% confidence level [46,47]. Residents who had been living in the selected TPUs for at least 24 months who were Chinese and aged 20 or above were invited to answer a questionnaire. Those who were cognitively impaired or unable to communicate effectively were excluded. Trained research assistants approached potential participants on the streets, in parks and outside the entrances of shopping malls in the TPUs. The participants were provided with an information sheet explaining the study and were reassured that their participation was voluntary and that the information provided was anonymous. Their specific names would not be associated with the reporting of the findings.

### 2.2. Assessments

#### 2.2.1. Quality of Life

The primary outcome of this study was QoL, which was measured using the WHOQOL-BREF (Hong Kong version). There were 26 items, all of which were rated using a five-point Likert scale. Cronbach’s alpha (α) for the scale ranged from 0.66 in the social domain to 0.84 in the psychological domain [4]. Confirmatory analyses showed that the comparative fit indexes ranged from 0.837 for the environmental domain to 1.0 for the social domain, demonstrating that the four domains had acceptable construct validity. The domain scores were calculated and transformed into a linear scale between 0 and 100, following the scoring guidelines [4]. A higher score indicated a better QoL. In the current study, Cronbach’s alpha coefficient for each domain of QoL was as follows: 0.74 for physical, 0.77 for psychological, 0.63 for social and 0.78 for environmental.

#### 2.2.2. Sociodemographic

Sociodemographic data included age, education level, occupation, marital status, monthly income, living location, housing types and others living in the same residence. For the socioeconomic status (SES), we categorised the participants based on their residential addresses with reference to the 2014/15 household expenditure survey in Hong Kong [48]. The participants were asked to provide the street name and number of their address for the team to denote the percentage of green space within a 500 m radius of their residence using a vegetation map of Hong Kong. This map was estimated based on object-oriented classification with high-resolution Satellite Pour I’Observation de la Terre (SPOT) image [49]. SPOT-6 satellite images consisted of panchromatic wavelengths at a 1.5 m spatial resolution and multispectral wavelengths at a 6 m spatial resolution. A digital elevation model (DEM) and digital surface models (DSMs) were used to separate vegetation structures from urban infrastructures and buildings. The vegetation map, derived by the project team through a contract from the Hong Kong Planning Department, displayed vegetation types (e.g., grassland, shrubland and woodland) and natural features (e.g., mangroves, badlands and rocky shores) [49,50].

#### 2.2.3. Environmental Conditions

The environmental conditions included environmental quality, green space percentage and public space satisfaction. Participants were asked to rate environmental quality based on the six parameters of air quality, heat island intensity, noise, vegetation density, building height, building density and overall environmental quality. Each parameter was illustrated with a local reference photograph. Participants were asked to rate how they perceived their neighbourhood environment for each parameter on a scale ranging from 0 = extremely low to 100 = high quality. The parameters were validated as urban environmental quality indicators [51]. In the current study, Cronbach’s alpha for environmental quality was 0.83. In addition, six questions on satisfaction with public spaces, including greening, parks and gardens, recreation and sports facilities, promenades and rest areas, were rated using a five-point Likert scale (1 = very unsatisfied to 5 = very satisfied), where Cronbach’s alpha was 0.83.

#### 2.2.4. Individual Health Behaviours and Health Utilisation

Individual health behaviours, including healthy eating, physical activity, smoking, alcohol consumption, sleep and perceived stress, were measured in the study. Healthy eating was measured by healthy eating behaviours and fruit and vegetable consumption. Fruit and vegetable consumption was defined using the WHO [52] recommendation of at least two portions of fruit and three portions of vegetables per day. Healthy eating behaviour was based on the recommendations of the Department of Health, Hong Kong [53], and was defined as a diet low in fat, salt and sugar, as measured using a four-point Likert scale (1 = never to 4 = always). Physical activity (PA) in the last 7 days was measured using the Chinese version of the International Physical Activity Questionnaire (IPAQ-C). PA was categorised into low-, moderate- or vigorous-intensity activities, and the total metabolic equivalent of task (MET) minutes per week was assessed. MET is the energy expenditure of physical activities [54]. Cronbach’s alpha for the IPAQ-C was 0.79, demonstrating an acceptable internal consistency. The concurrent validity of the IPAQ-C and MET-minute/week (%CV ranged from 49–113) with a total %CV of 43% [55]. Each PA activity was calculated according to the IPAQ scoring guidelines [56]. Smoking habit was reported as never smoked or former/current smoker. Alcohol consumption in the past year was measured using the Alcohol Use Disorders Identification Test–Concise (AUDIT-C) [57] survey. The scores ranged from 0 to 40, with higher scores indicating alcohol dependence. Cronbach’s alpha for the AUDIT-C was 0.79. The sleep quality of participants over the previous 3 months was measured using the eight-item Sleep Quality Index (SQI) [58]. Participants were asked to report the time taken to fall asleep (≤10 min, 11–30 min or >30 min), how often they had difficulty falling asleep, if they woke up during the night, woke up too early, had a disturbed night sleep or insomnia (no, <3 days/week or 3–7 days/week), morning tiredness (rather or very alert, do not know or rather or very tired) and their use of hypnotics (no, occasionally or at least 1 per week). The scores ranged from 0–16, with higher scores indicating more severe sleep disturbance [59]. Cronbach’s alpha for the SQI was 0.65 [60]. Perceived stress, including the ability to cope with stressors and the degree of negative emotional reaction towards stressors in the past month, was measured on a 10-item Perceived Stress Scale (PSS-10) using a five-point Likert scale [61]. The total score ranged from 0 to 40, with higher scores indicating higher perceived stress. The internal consistency of PSS–10 was good with a Cronbach’s alpha of 0.83 [62]. For health utilisation, questions on the regular use of Western and Chinese medications, doctor consultations for Western and Chinese medicine, hospitalisations and sick leave taken in the past three months were assessed.

## 3. Data Analysis

The data analysis was conducted using SPSS version 25 (SPSS Inc., Chicago, IL, USA), and the mediation analysis was implemented using the SPSS macro PROCESS (version 3) model 4 [62]. PROCESS is a computational tool that uses the bootstrap resampling method to provide a bias-corrected indirect effect. We specified 5000 bootstrap samples based on 95% confidence intervals (CIs). An indirect effect can be found when the 95% CIs do not include zero. The mediation analysis investigated whether both path a (predictor to mediator) and path b (mediator to the outcome) were significant, even if path c (direct effect of predictor to outcome) was insignificant [63]. All alphas were set at 0.05 (two-tailed).

Individual health behaviours regressed on environmental conditions were together estimated as path a. WHOQOL-BREF regressed on individual health behaviours were together estimated as path b. The direct effect of the environmental conditions on WHOQOL-BREF was together estimated as path c. To compare the influence of the variables between different domains in the WHOQOL, four separate models for four WHOQOL-BREF domains (physical—model 1, psychological—model 2, social—model 3 and environmental—model 4) were conducted in the study. Covariates to be included in the model were determined by the univariate linear regression. Only those variables reaching a significance level for the WHOQOL-BREF were included as covariates. Based on the previous research showing that there was a correlation between sociodemographic, individual health behaviours and QoL, we hypothesised that environmental conditions affect the QoL of adults. Covariates were used to control for the initial differences in QoL.

## 4. Results

### 4.1. Descriptive Characteristics of Participants

We recruited 614 participants to take part in the study, though seven questionnaires were incomplete and thus discarded. Of the eligible 607 participants, 390 were female and 217 were male, and their mean age was 47.4 years (SD = 21.1). Over 44% had attained university education and 59.8% were married. The percentage of the consumer price indexes (CPIs), in ascending order, was 48.6 (grade A), 38.7 (grade B) and 12.7 (grade C). Over 63% of the participants said that their health was good or very good and 35.3% reported living with chronic illnesses. In terms of their health utilisation over the past 3 months, 55% of participants sought doctor consultations, 41.8% took regular medication, 9.1% took sick leave and 2.8% had been hospitalised. In terms of individual health behaviours, over 90% of the participants were non-smokers. The mean score for alcohol consumption was 1.0 ± 1.6. The majority of them reported that they did not meet the guideline for fruit (84.7%) and vegetable (90.4%) consumption. Over the previous 3 months, the mean MET-minutes/week was 2406.5 ± 1709.4, perceived stress was 15.9 ± 5.5 and sleep was 4.6 ± 3.4. For environment conditions, the mean score for green space was 10.1 ± 7.9. The perceived environmental quality was 57.7 ± 15.5, and public space satisfaction was 3.3 ± 0.7. For QoL, the mean score for the four domains were 60.5 ± 10.5 (physical), 62.8 ± 13.6 (psychological), 62.9 ± 12.5 (social) and 61.9 ± 13.5 (environmental). Table 1 gives the descriptive characteristics of the participants.

### 4.2. QoL and the Predictor Variables

The basic model for the demographic characteristics, individual health behaviours and environmental conditions contained all the covariates in the study. Those included in the model were determined by the univariate linear regression. Only those variables reaching similar significant levels as the WHOQOL-BREF were included as covariates. Males (*p* < 0.05) and individuals with chronic illness (*p* < 0.05) were significantly negatively associated with social QoL. A higher environmental QoL was reported among those who had attained higher education (*p* < 0.01) and who met the guidelines for fruit and vegetable consumption (*p* < 0.05). Participants who met the guidelines for both fruit and vegetable consumption had higher QoL. Table 2 provides the covariates.

### 4.3. Mediation Effects for Physical QoL

The regression model for physical QoL was significant: F (8, 598) = 41.08, *p* < 0.001, R^2^ = 0.36. The bootstrap result revealed that perceived stress was a mediator of environmental quality (β = 0.046, 95% CI = 0.003–0.072) and public space satisfaction (β = 0.056, 95% CI = 0.002–0.112) in model 1. Sleep quality only mediated environmental quality (β = 0.033, 95% CI = 0.013–0.055). Fruit and vegetable consumption was adjusted to test whether physical QoL improved when environmental conditions were introduced after the mediators had been included. Figure 1 shows the mediation model for physical QoL.

### 4.4. Mediation Effects for Psychological QoL

The regression model for psychological QoL was significant: F (8, 598) = 43.50, *p* < 0.001, R^2^ = 0.37. The bootstrap result revealed that perceived stress was a mediator of environmental quality (β = 0.067, 95% CI = 0.034–0.102) and public space satisfaction (β = 0.081, 95% CI = 0.004–0.165). Sleep quality only mediated environmental quality (β = 0.032, 95% CI = 0.012–0.057). Figure 2 shows the mediation model for psychological QoL.

### 4.5. Mediation Effects for Social QoL

After adjusting for fruit and vegetable consumption, gender and chronic illness, the overall regression model for social QoL was significant: F (10, 596) = 12.77, *p* < 0.001, R^2^ = 0.186. The bootstrap result revealed that environmental quality was mediated by perceived stress (β = 0.044, 95% CI = 0.021–0.071) and sleep quality (β = 0.021, 95% CI = 0.007–0.040). Figure 3 shows the mediation model for social QoL.

### 4.6. Mediation Effects for Environmental QoL

After adjusting for fruit intake and education level, the regression model for environmental QoL was significant: F (10, 596) = 38.54, *p* < 0.001, R^2^ = 0.39. The bootstrap result revealed that environmental quality was mediated by perceived stress (β = 0.046, 95% CI = 0.022–0.075) and sleep quality (β = 0.021, 95% CI = 0.007–0.039). Figure 4 shows the mediation model for environmental QoL.

## 5. Discussion

Quality of life models often focus on dimensions of physical and psychological health. Most of the literature investigating QoL does not specify the links between environmental conditions and QoL domains of health. This study demonstrated that environmental conditions do affect QoL, and as a core element of QoL, it is linked with various factors. In this study, both objective (green space percentage) and subjective (environmental quality and public space satisfaction) aspects of the environmental conditions affecting QoL were analysed.

The studied population resided in a typical urban neighbourhood covered by limited green space. The level of green space in the study districts was low (mean 10.1 ± 7.9, range 0–100), as is the case in most urban areas of Hong Kong. The findings suggest that it positively affected physical activity and physical QoL, even if the space was limited. Previous studies suggest that green space has a positive effect not only on physical health and activity [25,30,44,64] but also on psychological health [21,28,65]. In our study, there were significant benefits of green space on physical activity and physical QoL but not on psychological QoL. This may be due to factors such as perceived stress mediating the results. Studies suggested the exposure to green space is a major factor that affects perceived stress. Every 1% increase in exposure to green space is associated with decreased cortisol levels and decreased self-reported stress [33]. Living less than 1000 m from a green space can lead to less stress and improved QoL [11]. In our study, we measured the green space cover within a 500 m radius of residence and found no benefit to psychological health and stress.

Although the objective value of green space in the study districts was low, the satisfaction with public space was moderate (mean 3.3 ± 0.7, range 0–5) in our study. Public space satisfaction was positively associated with all domains of QoL. This result was similar to those of other studies on public space satisfaction [64,66]. Public space satisfaction refers to leisure and recreation facilities. Although there is no standard for satisfactory public spaces, the requirements of the residents in the studied districts were met. People reporting less satisfaction with public spaces had higher perceived stress. As a mediating factor, the higher stress negatively affected both physical and psychological QoL. Perceived stress was related more strongly to the quality of public spaces than the quantity [26]. A local study suggested that recreational and public spaces are important in Hong Kong for environmental satisfaction, regarding both private and public housing [24]. Housing environments in neighbourhoods are fundamental to residents’ overall life satisfaction [67], but the expectations for their housing environments are inconsistent. Those living in public housing were concerned about the accessibility of the housing location, whereas those in private and subsidised housing were more concerned with the appearance of the housing environment. The association of QoL with the types of housing and their locations should be further studied.

Besides green and public space satisfaction, environmental quality also influenced QoL and mediated both stress and sleep. In this study, environmental quality was negatively associated with QoL through perceived stress and poor sleep. Living in an urban neighbourhood can be stressful, where exposure to environmental risks, such as heavy traffic and pollution, can negatively influence general health [68]. A significant association has been found between perceived stress and physical exposure to noise, the safety of public spaces and QoL [69]. The detrimental effect of stress on health is evident. In our study, perceived stress was shown to have a significantly negative effect on QoL (β = 0.622–0.961). Sleep was also a significant mediator of all four QoL domains in this study; as a physiological process that restores and repairs body functions, good sleep is fundamental to health and quality of life. Like perceived stress, poor sleep quality had a significant negative effect on QoL (β = 0.566–0.890). This finding was in accordance with previous research, which showed that people with poor sleep perceived lower QoL [37].

Stress and poor sleep are common risk factors of non-communicable diseases. The local government launched a strategic plan for 2025, involving multi-sectoral actions, including urban planning and the creation of health-supporting environments to enable people to make healthy choices, to enjoy healthy living and to fight against non-communicable diseases [70]. Various measures can be taken immediately to reduce the disease burden that is attributable to environmental determinants. The government should consider taking not only access to green space but also the quality of the environment and public space into consideration when designing healthy communities. These can support good neighbourhood environments and the livelihood of people. The findings suggested that poor sleep is associated with perceived higher levels of stress and increased reports of physical health discomfort and poor mental QoL. Strategies to improve environmental quality are essential to reduce stress and promote sleep, thus resulting in good QoL.

Urbanisation contributes to the fact that more than half of the world’s population lives in cities [71]. There has been a growth of research interest regarding the impact that the urban environment has on health and QoL through its physical design, land use and service delivery [72]. The impacts of the urban environment on human health have been targeted for public health policy across the globe. The importance of promoting a healthy environment in both developed and developing countries and region, such as Africa Australia, China and England, have been reinforced [73]. International objectives and strategies to enhance the environment and health have been emphasised amongst international organisations, including the World Health Organization [6], the World Bank [74], the European Union [75] and the United Nations [76]. Previous studies have observed the impact of the environment on health; for example, the influence of neighbourhood safety, pollution and traffic on physical and psychological well-being. Environmental conditions can influence residents’ capability to handle stress and maintain mental health [77]. Stress can be a positive or negative life process [78]. Urban environment conditions are both a source and a resource that help people to cope with stress [79] and to get proper sleep [40,80].

This study explained the dynamics of QoL in the urban city. The model helped to describe and determine the causal relationships between environmental conditions and QoL, with the mediating effects of stress and sleep. Our study has several limitations. First, the limited sample size and homogenous culture of the participants may affect the generalisability of the results. A more diverse culture and larger samples could be considered to further confirm the association between environmental quality and QoL. Second, the outcomes of the study were based on subjective measures. Future studies could include objective measurements, such as those of cortisol levels for stress and physical fitness tests. Associations by age category and different neighbourhood environments of the participants could further be explored in future studies.

## 6. Conclusions

In this study, WHOQOL-BREF served as an important outcome regarding how residents interact with their neighbourhood environment. The findings suggest that access to green space was associated with physical QoL and physical activity. Individuals with a positive evaluation of public space satisfaction and environmental quality were more likely to perceive a good quality of life. The influence of objective and subjective measures of environmental conditions on QoL was mediated by stress and sleep. The findings of this study further supported the importance of individuals’ stress and sleep in mediating the relationship between the environment and, as a direct consequence, QoL. Environmental conditions and interventions should be considered whenever making public health policies regarding urban populations.

## Figures and Tables

**Figure 1 ijerph-17-08529-f001:**
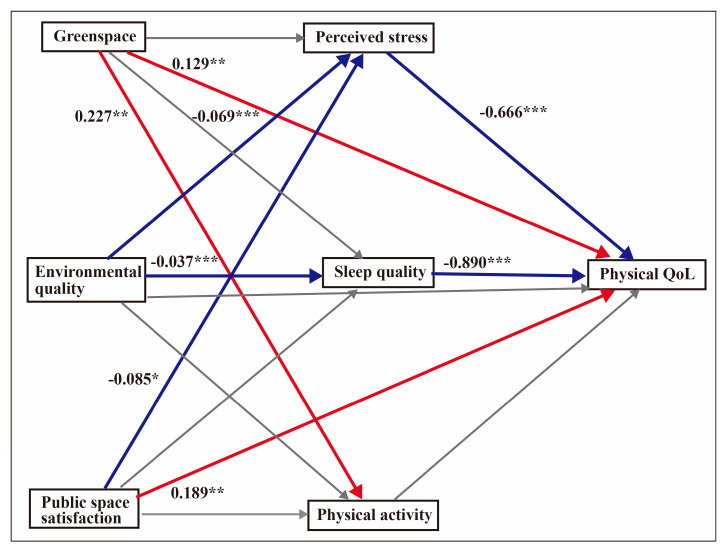
Mediation model for physical QoL. The analyses controlled for fruit intake and vegetable intake. Unstandardised coefficients are displayed. The values indicate the strength of the relationship between the variables. Red arrows depict positive relationships, blue arrows show negative relationships and grey arrows depict non-significance. * *p* < 0.05, ** *p* < 0.01, *** *p* < 0.001.

**Figure 2 ijerph-17-08529-f002:**
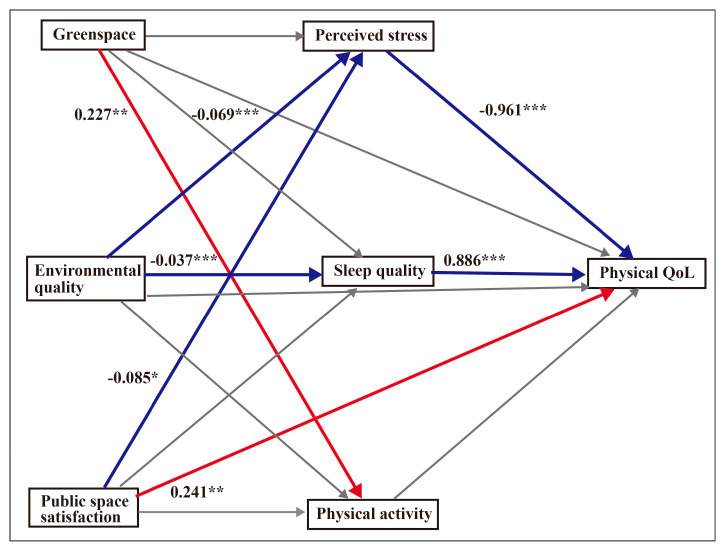
Mediation model for psychological QoL. The analyses controlled for fruit intake and vegetable intake. Unstandardised coefficients are displayed. The values indicate the strength of the relationship between variables. Red arrows depict positive relationships, blue arrows show negative relationships and grey arrows depict non-significance. * *p* < 0.05, ** *p* < 0.01, *** *p* < 0.001.

**Figure 3 ijerph-17-08529-f003:**
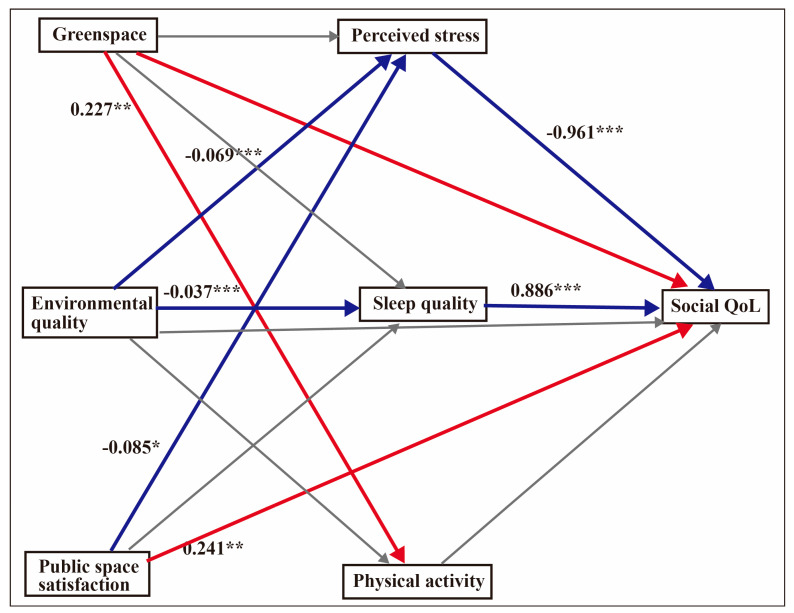
Mediation model for social QoL. The analyses controlled for fruit intake, vegetable intake, gender and chronic illness. Unstandardised coefficients are displayed. The values indicate the strength of the relationship between variables. Red arrows depict positive relationships, blue arrows show negative relationships and grey arrows depict non-significance. * *p* < 0.05, ** *p* < 0.01, *** *p* < 0.001.

**Figure 4 ijerph-17-08529-f004:**
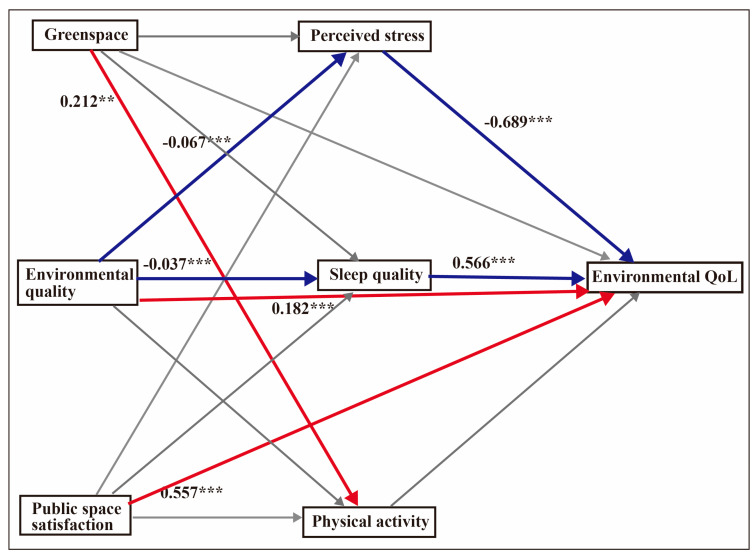
Mediation model for environmental QoL. The analyses controlled for fruit intake and education level. Unstandardised coefficients are displayed. The values indicate the strength of the relationship between variables. Red arrows depict positive relationships, blue arrows show negative relationships and grey arrows depict non-significance. ** *p* < 0.01, *** *p* < 0.001.

**Table 1 ijerph-17-08529-t001:** Descriptive characteristics.

Variables	*n* = 607
Age, M ± SD (years)	47.4 ± 21.1
Gender, *n* (%)	
Female	390 (64.3)
Male	217 (35.7)
Education, *n* (%)	
Primary	117 (19.3)
Secondary	2178 (29.3)
Diploma	42 (6.9)
University	270 (44.5)
Marital status, *n* (%)	
Single	192 (31.6)
Married	363 (59.8)
Widowed/divorced/separation	52 (8.6)
Occupation, *n* (%)	
Not working	202 33.3)
Working	193 (31.8)
Homemaker or student	212 (34.9)
AUDIT-C (0–12), M ± SD	1.0 ± 1.6
Smoking status	
Non-smoker	552 (90.9)
Some days	22 (3.6)
Every day	33 (5.4)
Fruit intake, *n* (%)	
Did not meet the guideline	514 (84.7)
Met the guideline	93 (15.3)
Vegetable intake, *n* (%)	
Did not meet the guideline	549 (90.4)
Met the guideline	58 (9.6)
Chronic illness, *n* (%)	
No	393 (64.7)
Yes	214 (35.3)
CPI, *n* (%)	
Grade A (5500–24,499)	295 (48.6)
Grade B (24,500–44,499)	235 (38.7)
Grade C (44,500–89,999)	77 (12.7)
Individual monthly income, *n* (%)	
0	259 (42.7)
<10,000	143 (23.6)
10,000–14,800	59 (9.7)
14,800–23,000	51 (8.4)
>23,000	95 (15.7)
Green space (0–100), M ± SD	10.1 ± 7.9
Environmental quality (0–100), M ± SD	57.7 ± 15.5
Public space satisfaction (0–5), M ± SD	3.3 ± 0.7
Sleep Quality Index (0–14), M ± SD	4.6 ± 3.4
IPAQ, M ± SD	2406.5 ± 1709.4
Perceived Stress Scale (0–40), M ± SD	15.9 ± 5.5
WHOQOL (0–100), M ± SD	
Physical	60.5 ± 10.5
Psychological	62.8 ± 13.6
Social	62.9 ± 12.5
Environmental	61.9 ± 13.5
Doctor, *n* (%)	
No	273 (45.0)
Yes	334 (55.0)
On medication, *n* (%)	
No	353 (58.2)
Yes	254 (41.8)
Hospitalisation, *n* (%)	
No	590 (97.2)
Yes	17 (2.8)
Sick leave, *n* (%)	
No	552 (90.9)
Yes	55 (9.1)

CPI: Consumer Price Index, AUDIT-C: Alcohol Use Disorders Identification Test, IPAQ: The International Physical Activity Questionnaire–Short Form, WHOQOL: World Health Organization Quality of Life 100.

**Table 2 ijerph-17-08529-t002:** Univariate analysis (linear regression).

Variables	Physical QoL	Psychological QoL	Social QoL	Environmental QoL
Age	0.051	0.010	−0.047	0.060
Gender				
Female	Ref	Ref	Ref	Ref
Male	0.048	0.027	−0.090 *	−0.13
Education				
Primary	Ref	Ref	Ref	Ref
Secondary	0.008	−0.494	−0.490	1.298 **
Diploma	−0.098	−0.265	−0.326	0.792 **
University	−0.014	−0.415	−0.419	1.510 **
Marital status				
Single	Ref	Ref	Ref	Ref
Married	0.089	0.069	0.057	0.019
Widowed/divorced/separation	−0.006	−0.023	0.005	0.002
Occupation				
Not working	Ref	Ref	Ref	Ref
Working	−0.079	−0.024	−0.003	−0.132 **
Homemaker or student	−0.037	0.023	0.036	−0.060
AUDIT–C (0–12)	−0.006	−0.076	−0.009	−0.030
Smoking status				
Non-smoker	Ref	Ref	Ref	Ref
Some days	−0.059	−0.068	−0.071	−0.045
Every day	−0.065	−0.045	−0.059	−0.062
Fruit intake				
Did not meet the guideline	Ref	Ref	Ref	Ref
Met the guideline	0.085 *	0.130 **	0.092 *	0.087 *
Vegetable intake				
Did not meet the guideline	Ref	Ref	Ref	Ref
Met the guideline	0.122 **	0.089 **	0.090*	0.039
Chronic illness				
No	Ref	Ref	Ref	Ref
Yes	−0.028	−0.055	−0.090 *	−0.009
CPI				
Grade A (5500–24,499)	Ref	Ref	Ref	Ref
Grade B (24,500–44,499)	0.036	0.072	0.070	0.190 *
Grade C (44,500–89,999)	−0.063	−0.076	−0.006	0.074
Individual monthly income				
0	Ref	Ref	Ref	Ref
<10,000	−0.001	0.022	0.023	0.008
10,000–14,800	−0.050	−0.044	−0.035	−0.079
14,800–23,000	0.010	−0.001	0.022	0.019
>23,000	−0.060	0.027	0.043	−0.006
Greenspace (0–100)				
Environmental quality (0–100)	0.062	−0.012	−0.048	−0.027
Public space satisfaction (0–5)	0.270 ***	0.318 ***	0.150 ***	0.451 ***
Sleep Quality Index (0–14)	0.269 ***	0.291 ***	0.183 ***	0.449 ***
IPAQ	−0.478 ***	−0.361 ***	−0.251 ***	−0.296 ***
Perceived Stress Scale (0–40)	0.116 **	0.121 **	0.034	0.034
Doctor	−0.407 ***	−0.521 ***	−0.334 ***	−0.387 ***
No				
Yes	Ref	Ref	Ref	Ref
On medication	−0.025	−0.025	−0.037	0.050
No				
Yes	Ref	Ref	Ref	Ref
Hospitalisation	−0.053	−0.058	−0.061	0.047
No				
Yes	Ref	Ref	Ref	Ref
Sick leave	−0.075	−0.012	0.002	−0.028
No				
Yes	Ref	Ref	Ref	Ref
	−0.028	0.021	0.082	−0.011

* *p* < 0.05, ** *p* < 0.01, *** *p* < 0.001. QoL: quality of life, CPI: Consumer Price Index, AUDIT-C: Alcohol Use Disorders Identification Test, IPAQ: The International Physical Activity Questionnaire–Short Form, WHOQOL: World Health Organization Quality of Life 100.

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
