# Peer review of "The Impact of the Environment on the Quality of Life and the Mediating Effects of Sleep and Stress"

_ijerph, 2020, doi:10.3390/ijerph17228529_

Round 1
Reviewer 1 Report
Thank you for a chance to review this article. This is an interesting study.
Introduction is well-written, but I would suggest extending it a bit by adding more information from research on issues like 'city noise' or green city policy (especially coming from the traffic). Looking into a work "Neigborhood influences on physical activity in middle-aged and older adults: a multilevel perspective" could give authors some broader perspective. Also sleep deprivation could be better presented in the Introduction - not just in case of polluted, busy cities (like Tokio, Rio de Janeiro, New York or London), but also concernig similar weather climate (Jakarta, Singapour?)
In the Methods section authors provide a discription of the selection process, which seems fair and reasonably desinged, however, they also provide a minimum age of the participant but do not say anything about the maximum age, and this is important information. While for a 20-years old sleeping requirement (and physical ability to deal with potential sleep disorders or depravation) would be easy to handle, this may come more difficult for a 50-year old, for example. So I think mean age figure with SD should be reported in this subsection describing participants. However, this is provided later in the section Results, but I don't think this should be counted as results, but as a description of the research sample. Also here one thought - including in a sample size participants with age of 20 and analyzing them in the same manner as 60 years olds may cast some doubts. They needs, habits, material status, life and health conditions and experiences may differ much. Maybe it is better to narrow the sample size to age homogenius group?
Although, using IPAQ for accessing PA is always questionable, but the rest of the research tools do not ring any worrying bells, authors provide all the coefficients and this seems fine.
In the Results authors report that higher QoL was associated with higher education - what about those 20-years olds who took part in the research sample - they haven't gained their higher education qualification yet - they are probably treated as students, but their educational status/level may have disturbed the statistical analyzes. I also find interesting this negative relation of sleep and physical activity (fig.1) - how do you explain that?
In the final sections of the paper authors mainly discuss their own findings writting that generalization of the results may be difficult as the group was very much homogenious. And this is one of the shortcomings of the study - it local character. Perhaps more references to the situation of sleep, PA, health of citizens in other world metropolies would help to place this study in a broader context.
The paper seems as a valuable contribution to the field, but I suggest authors revise the paper and work more on the internationalizing the rationale for the study.
Author Response
The file is in the attachment

Reviewer 2 Report
Abstract
I am not sure what is the rationale for this study? The first two sentences are hard to read, and the entire abstract should be re-written for better flow.
The results are presented in a vague manner: what are physical and psychological factors? There is no need to list covariates in the abstract.
A good abstract must tell the reader: why a study was done; how was it done, what was found and what the findings mean.
Introduction
I was not entirely sure what is the focus of 2nd paragraph on p. 2 lines 64 – 87. SES, health practices (behaviours would be a more intuitive term) and stress are mentioned, yet not all concepts received a similar attention, in particular stress. I’m not sure it fits into this para unless it is elaborated in more detail.
Stress and sleep feature in the title yet do not receive an adequate attention in the intro.
Methods and Analysis
The methods section would benefit from more headings.
Please provide Cronbach alpha for all questionnaires used in the study.
What were the covariates used, and what was the rationale for their inclusion?
Results
Please introduce headings and subheadings. Otherwise clear.
Discussion
What were the study strengths and limitations?
Line 287: “In our study, there were significant benefits of green space on physical activity and physical health but not on psychological health.” – it is not clear, based on your method section, how you measured psychological health in this study.
Line 289: “Every 1% increase in exposure
to green space is associated with an increased cortisol level and decreased self-reported stress.” It should be decreased cortisol.
Line 299 – this sentence is not very clear: “On the contrary, people reported less satisfaction with public spaces were negatively affects both physical and psychological QoL by perceived stress.”
You need to more clearly explain what the established mediations actually mean, in particular, stress and sleep feature in the title yet do not receive an adequate attention in the discussion.
Author Response
The file is in the attachment

Round 2
Reviewer 1 Report
Text has been much modified accordingly to the suggestioins of the reviewers,
and now it meets the standards of IJERPH
Reviewer 2 Report
Thank you for addressing my comments.
I think the abstract is far more comprehensive now, but please edit the entire paper in terms of English language expression.
I still want to see more discussion about sleep and stress in paragraph 5, page 3. Also, why is reaction time relevant here (this is a newly added sentence by authors):
Environment features such as light, noise, traffic can affect perceived sleep quality
Chronic exposure to an environmental noise by road traffic can decrease perceived sleep quality,
mood and performance in terms of reaction time [35].
You can just elaborate the findings you already cite, no need to add more studies.
